# Analysis of the Microstructure and Porosity of Cement Pastes with Functionalized Nanosilica with Different Contents of Aminosilane

**DOI:** 10.3390/ma16165675

**Published:** 2023-08-18

**Authors:** Gabriel Lima Oliveira Martins, Yuri Sotero Bomfim Fraga, Andréia de Paula, João Henrique da Silva Rêgo, Amparo Moragues Terrades, Moisés Frías Rojas

**Affiliations:** 1Department of Civil and Environmental Construction, University of Brasília, Brasilia 70910-900, Brazil; gabriel.martins30@outlook.com (G.L.O.M.); yuri.fraga@ufac.br (Y.S.B.F.); paula_adp@yahoo.com.br (A.d.P.); jhenriquerego@unb.br (J.H.d.S.R.); 2Center for Exact and Technological Sciences, Federal University of Acre, Rio Branco 69920-900, Brazil; 3Department of Construction Materials, Universidad Politécnica de Madrid, 28040 Madrid, Spain; amparo.moragues@upm.es; 4Eduardo Torroja Institute for Construction Science, 28033 Madrid, Spain

**Keywords:** Portland cement, hydration kinetics, functionalized nanosilica, nanotechnology

## Abstract

This research aims to analyze the effect of functionalized nanosilica (NSF) with different levels of amine groups in the formation of hydration products. Four cement pastes were investigated, one reference with Portland cement and three replacing 1% of Portland cement by nanosilica (NS), NSF with a low content of amine groups, and NSF with a high content of amine groups. The heat of hydration of the pastes was evaluated up to 7 days of hydration, the amount of calcium hydroxide (CH) and hydrated phases by means of the thermogravimetric analysis (TGA) test and compressive strength at 2, 7, and 28 days, and porosity through tests of mercury intrusion porosimetry and computed tomography at 28 days of hydration. It was possible to observe that the NSF directly influenced the hydration kinetics of the pastes, delaying the hydration of the Portland cement; however, it demonstrated a similar mechanical performance to the paste with NS at 2 days of hydration and an increase of 10% at 28 days of hydration due to the improvement in the hydration process. Thus, it is possible to conclude that the functionalization of NSF with a low 3-aminopropyltriethoxysilane (APTES) content is promising for use in cementitious materials and may improve hydration and mechanical performance at more advanced ages compared to NS.

## 1. Introduction

The use of pozzolanic materials is beneficial for cementitious composites [1,2]. The use of nanosilica (NS) as a supplementary nano cementitious material (SNCM) is a consolidated practice in the scientific community for the production of high-performance cementitious materials [3,4,5,6,7]. The combination of the high specific surface of NS and more efficient pozzolanic activity promotes a denser nano/microstructure that justifies the increase in compressive strength and the improvement in the durability of cementitious products [5].

Despite the potential to improve the properties of cementitious composites, NS still has open fields of study and adverse effects in the fresh and hardened state that can be solved with technological advances. In the fresh state of cementitious composites, Gu et al. [8] highlighted the adversity caused by the tendency for agglomeration between silica nanoparticles. In order to obtain a better composite performance, the nanoparticles need to be dispersed in the matrix, so a dispersion procedure is necessary in advance [6].

In the hardened state, the use of NS is commonly associated with a decrease in strength gain compared to pure cement at advanced ages. Some authors attribute this drop in strength potential at older ages to the effect of the incomplete hydration of clinker grains [9,10,11,12]. This incomplete hydration of the largest clinker grains occurs due to the formation of a C-S-H crust on its surface by the intense pozzolanic reaction of NS in the first hours of cement hydration, which prevents the complete hydration of these grains at more advanced ages. To correct and/or mitigate these adverse effects, some authors have suggested altering the surface of the NS by functionalization processes [13,14].

Functionalization is a process that consists of adding new chemical functions on the surface of the NS. This chemical reaction replaces the silanol groups (OH^−^) on the NS surface by another function of higher interest [14,15,16,17,18]. One of the functional groups that have stood out the most for the functionalization of NS is the aminosilanes groups, forming a new material: nanosilica functionalized with aminosilanes groups (NSF) [11,19,20]. The NSF production process by functionalization consists of a chemical reaction that replaces the silanol groups (OH^−^) on the surface of the NS by aminosilane groups, in this case 3-aminopropyltriethoxysilane (APTES), directly.

According to Khalil et al. [21], amine groups are known to have polarity that allows them to disperse in ionic media (such as Portland cement pastes) more easily than other organic groups, such as the silanol groups present in NS. Collodetti et al. [11] and Vasconcelos et al. [12] observed the need for a lower superplasticizer content in cement pastes with NSF compared to cement pastes with NS to achieve the same consistency, which confirms the higher dispersion of NSF in cementitious medium. The same authors observed a delay in cement hydration with a consequent reduction in compressive strength at initial ages (1 and 3 days) but an increase in compressive strength at more advanced ages (28 days) in pastes with NSF when compared to pastes with NS. Rong et al. [20] studied NSF with different contents of amine groups (KH550) and, based on the initial characterizations of this material, proceeded with the study of only one content in cement pastes. The selected sample of NSF, with 10% of KH550, showed an improvement in the compressive strength at ages of 3, 7, and 28 days in relation to the reference sample (OPC).

The nanosilica functionalization process with different aminosilane contents (APTES) can have three possible functionalized structures, as shown in Figure 1. The first structure is shown in Figure 1a, which represents a conventional NS, with OH^−^ groups on the surface. In Figure 1b, it is possible to verify that the NSF has a small amount of APTES linked to the OH^−^ groups, but with free OH^−^ groups, favoring the pozzolanic reaction. In Figure 1c, it is possible to observe that most of the OH^−^ groups were linked to APTES, saturated NSF, including due to the high APTES content, and that the silane group was linked to the aminosilane group, forming a double layer of APTES on the surface of the NSF.

This alteration in the relationship between OH^−^/APTES groups on the surface of the NS can change the behavior of the NSF when incorporated into the cementitious material. Thus, this research aims to evaluate how the functionalized NSF with a low APTES content (Figure 1b) and the functionalized NSF with a high APTES content, forming a double layer of functionalization (Figure 1c), influence the hydration of pastes of cement and, in particular, how these NSFs interfere in the compressive strength and porosity of these pastes.

## 2. Materials and Methods

The Portland cement used was CP-V ARI Portland cement in accordance with NBR 16697. This cement is also named by ASTM C150-5 as type III cement.

Colloidal nanosilica (NS), used as a raw material for NSF and the reference sample, has nanosilica in suspension, produced by AkzoNobel. Functionalized nanosilica synthesis procedures (NSF) were demonstrated in a previous research [12].

The NSF samples were named according to the amount of APTES in milliliters that was added to the 60 mL of colloidal nanosilica, as listed below:NSF4—containing 4 mL of APTES (NSF low in APTES);NSF8—containing 8 mL of APTES (NSF high in APTES).

The nanosilica and functionalized nanosilica samples were generally characterized by their dispersion color, specific mass, pH, loss on ignition, and solids content. In addition, the following techniques were used to characterize NS, NSF4, and NSF8: dynamic light scattering (DLS), thermogravimetric analysis (TGA), and infrared spectroscopy (FT-IR).

The size distribution of the nanosilica particles was determined using the Zetasizer Nano ZS90 device (Malvern Panalytical, Malvern, UK), which determines the size by the hydrodynamic diameter and has already been used in other similar studies for the same purpose [22,23,24].

To quantify the content of functionalized amine groups on the NS surface, thermogravimetric analysis (TGA) was used with the Q600 model device (TA Instruments, New Castle, DE, USA) operating at 10 °C/min up to a temperature of 1000 °C with nitrogen flow. From this analysis, the functionalization content is obtained, which was taken as the subtraction of the sample’s total mass loss minus the sample’s total loss of NS.

The identification of chemical bonds of functionalization was verified by changing the vibrational modes present on the surface of the NS by the spectra generated in infrared spectroscopy (FTIR). The equipment used was Perkin Elmer FT-IR, Spectrum 400 (Waltham, MA, USA).

Four samples of Portland cement pastes were produced with a water/solid ratio of 0.35, as shown in second table in Section 3.1 The superplasticizer additive content was determined by setting the spread of the paste at 94 ± 4 mm in the mini slump test described by Kantro [25]. The amount of water was adjusted considering the liquid/solid content discounts of the materials used. The composition of the pastes is shown in Table 1.

The paste mixture process followed the recommendations of NBR 5739, with adaptations. Water, superplasticizer additive, nanosilica (NS, NSF4, and NSF8), and Portland cement were added, respectively, in a stainless steel vat. After the cement came into contact with the water, 30 s of rest was counted, then the planetary mixer was turned on at a low speed (140 rpm rotation around the axis). It was turned off for 60 s; during the first 30 s, the internal walls of the vat were scraped with a rubber spatula. After this process, the mixer was turned on at a high speed (285 rpm rotation around the shaft).

It is observed that a larger amount of superplasticizer additive was necessary to reach the desired consistency in the pastes with the NS and in the NSFs in relation to the reference paste. The reference paste required 0.28% SP and the NS paste required 0.80%. According to Lavergne et al. [26], the addition of nanosilica in the cement paste and mortar requires more water to maintain its workability, and the reason for this is attributed to a decrease in the available amount of lubricating water in the mixture. For the P-NSF4 sample, there was a decrease in the superplasticizer content in relation to the NS paste. This decrease may be related to two factors: the first is the slight increase in particle size from NS to NSF4 (from 21.74 nm to 26.04 nm) decreasing the surface area for water adsorption, and the second factor is the chemical affinity that the aminosilane group has with the polycarboxylate-based additive. This same chemical affinity is reported by Vasconcellos et al. [12] to justify the decrease in the additive content in cementitious composites with NSF. However, the P-NSF8 paste required more superplasticizer additive than the other pastes, which can be explained by the effect of water retention in the NSF8 particle, that is, part of the water can be encapsulated within the branches generated by the functionalization similar to a superabsorbent polymer, as evidenced in the literature [27].

After molding the pastes into cylindrical specimens, the samples were cured in a humid chamber (humidity of 95 ± 5% and temperature of 20 ± 2 °C) and ruptured. The specimens, which were 50 mm in diameter and 100 mm in height, were broken to verify the compressive strength at 2, 7, and 28 days of hydration. The internal fragments of the samples at the tested ages underwent a hydration stoppage process with 24 h in isopropanol and, subsequently, 24 h in an oven at 40 °C [28,29].

Isothermal conduction calorimetry tests were carried out on the P-REF, P-NS, P-NSF4, and P-NSF8 cement pastes from the first hours to 7 days, using the eight-hour isothermal conduction calorimeter TAM AIR Thermometer channels with temperature control, manufactured by TA Instruments, and data acquisition was performed by PicoLog 6 software.

Thermal analyses were performed on all cement pastes studied at the ages of 2, 7, and 28 days. Thermogravimetric curves were obtained using a Shimadzu differential thermal and thermogravimetric analysis system, model DTG-60H, between room temperature and 1000 °C, at a heating rate of 10 °C/min and under nitrogen flow (50 mL/min). The mass of the samples used in the analysis ranged from 6 to 12 mg.

To verify the compressive strength of the pastes, three specimens of each mix were ground and broken in a universal machine for mechanical tests at each of the following ages: 2, 7, and 28 days. The breaking procedures followed NBR 7215 [30].

The MIP test was carried out on the pastes at 28 days of hydration. The equipment used for the test was the Micromeritics Poresizer, model 9320 (Norcross, GA, USA). A contact angle of 130 °C was used, mercury with a surface tension of 0.485 N/m, and a density of 13.5335 g/mL. The pressure range used in the test ranged from approximately 0.50 psi to 29,472.38 psi.

The CT-scan test was performed on the pastes at 28 days of hydration. X-ray computed tomography imaging (CT) was obtained on a Nikon XT-H-160 scanner (Tokyo, Japan), equipped with a 160 kV W target and a 0.75 mm Cu filter. In total, 3015 scans per sample were recorded, at 2 frames per scan.

## 3. Results and Discussions

The following results demonstrate the characterization of the functionalized nanosilica, the influence of NSF with a high level of functionalization (NSF8), and a low level (NSF4) of APTES/NS on the behavior of pastes in the fresh and hardened state.

### 3.1. Characterization of Functionalized Nanosilica

The color, specific mass, pH, loss on ignition, and solids content of nanosilica and functionalized nanosilica samples are shown in Table 2.

It was possible to observe that the higher the content of amine groups, the whiter the sample, indicating the functionalization of NSF4 and NSF8. Furthermore, the higher the degree of functionalization, the higher the loss on ignition and the lower the solids content of the samples. This was important to define the effective NSF content of the pastes. The DLS and TGA results are listed in Table 3.

It is observed in Table 3 that the NSF8 sample presented a higher level of functionalization (9.57%), while the NSF4 sample presented 3.22% of amine groups. An increase in the hydrodynamic diameter of the particles was observed with the increase in the level of functionalized APTES on the surface of NSF4 and NSF8. In NSF8, it can be observed that two characteristic regions were formed (A1 and A2), with particle sizes of 203.5 nm (39.3%) and 858.8 nm (56.7%). This increase in particle size accompanied by the increase in the percentage of the second area can characterize the formation of a multilayer system on the surface of the nanosilica, and therefore this sample can be a portrait of an NS with excess of APTES (Figure 1c). As for the NSF4 sample, the presence of the second particle distribution area has a significantly lower percentage than the NSF8 sample. This effect was idealized as a partially functionalized sample (Figure 1b).

Figure 2 shows the FTIR spectra of the APTES, NS, and NSF samples. In the spectra, it is possible to identify the vibrational modes associated with O-H in different types of silanols present on the surface of NS and NSF. In addition to the silanol groups, stretching of the C-N groups is observed in APTES at approximately 1120 cm^−1^ (Sigma-Aldrich; São Paulo, Brazil). Note in Figure 1 that the intense band near 1120 cm^−1^ is slightly upwards shifted from the peaking band around 1080 cm^−1^, and the latter is attributed to the Si-O-Si asymmetric elongation mode of NS. Furthermore, the peak close to 800 cm^−1^ is attributed to the symmetrical stretching modes of the Si-O-Si and C-H bonds, and the latter is associated with APTES [31], thus explaining its increase in all NSF samples. The peaks close to 3400 cm^−1^ and 1200 cm^−1^ are attributed to the stretching mode of the Si-OH bond that is associated with the Q2 and Q3 groups. It is noteworthy that the species Q2, Q3, and Q4 represent -Si(-O-Si)_2_(OH)_2_, -Si(-O-Si)_3_(OH), and -Si(-O-Si)_4_ bonds for silicon, respectively.

### 3.2. Microstructure and Porosity of Cement Pastes

#### 3.2.1. Isothermal Conduction Calorimetry

Within the scope of the calorimetry test, the samples were analyzed for two behaviors: the first by the maximum intensity of the flow (Figure 3) and the second by the time it took to induce the chemical reactions, that is, the beginning of the reactions (Figure 4).

Observing the shape of the heat flux curves, one can notice the appearance of ettringite (E) by a shoulder in the hydration curve right after the peak of silicates (C). This shoulder is observed in all pastes; however, in the paste with NSF8, the shoulder related to the formation of ettringite was more intense than that of the silicates. This possible late formation of ettringite may be linked to more pronounced hydration delays in the P-NSF8 sample. This report is also indicated by Vasconcellos et al. [12].

As for the intensities of the heat fluxes of the pastes, the highest intensity of heat flux (approximately 12 mW/g) was presented by the P-NS sample. Samples P-REF, P-NSF 4, and P-NSF 8 had similar maximum heat flux intensities.

Observing the dormancy period, that is, the period between time zero and the beginning of the curve of the samples, we have that the sample with NS presented the shortest period of induction followed by the P-REF sample (before 3 h). This effect is largely due to the potential of NS as a highly reactive supplemental nano cementitious material combined with the fineness of the CP V cement. As for the samples with NSFs, the induction time was characterized by a delay, which has also been observed in the literature [32]. In samples with NSF4, the delay was about 10 h, and for P-NSF8, the delay was more than 30 h. This effect can be explained by the presence of larger amounts of excess amine groups in the medium, functioning as a buffer at the beginning of cement hydration reactions. These delays in hydration reactions in NSFs are due to the poisoning effect of hydrate nucleation sites and the growth of hydrates at this site. This effect made the action of NSFs similar to what occurs with setting retardant additives and fourth generation superplasticizer additives [33].

#### 3.2.2. Thermogravimetry

Through TGA, mass losses were quantified in the temperature ranges that correspond to the decomposition of Ca(OH)_2_ (between 400 and 470 °C). The CH content was calculated according to Equation (1) [29,34,35]. The CH index was calculated by dividing the CH content of the analyzed paste (P-NS, P-NSF4, or P-NSF8) by the P-REF paste.
CH content = 4.11 × volatilized water content (1)

Table 4 shows the CH content and CH index in relation to the REF paste of the pastes at 2, 7, and 28 days of hydration.

At 2 days of hydration, it was possible to observe that the P-NS paste resulted in a lower CH index compared to the P-REF paste, which can be explained by the high reactivity of the nanosilica that promotes the pozzolanic reaction from the earliest ages. The P-NSF4 paste showed a larger amount of CH compared to the P-REF and P-NS pastes, which can be attributed to the moment of the nucleation effect of the cement by the nanoparticles, but the pozzolanic reaction did not yet occur. The P-NSF8 paste resulted in the lowest amount of CH due to the delay of hydration reactions due to the larger amount of APTES on the surface of the nanoparticles [36].

At 7 days of hydration, the P-NS paste, due to the pozzolanic reaction, had a lower CH content than the P-REF paste. The P-NSF4 paste resulted in a lower CH content than the P-REF paste, indicating the beginning of the pozzolanic reaction. On the other hand, due to the higher APTES content functionalizing NSF8, there was a delay in hydration reactions and, consequently, a lower CH content compared to other pastes [37].

At 28 days of hydration, the P-NS paste resulted in a smaller amount of CH compared to the P-REF paste due to the pozzolanic reaction. The P-NSF4 paste had a smaller amount of CH than the P-REF paste, but a larger amount of CH than the P-NS paste. One hypothesis for this behavior is that a layer of C-S-H is formed on the surface of larger clinker grains in pastes with NS, impairing their complete hydration, while in NSF4 and NSF8, clinker hydration is improved, producing a larger amount of CH [32].

According to Frías et al. [38], the ranging from 50 to 400 °C is associated with the dihydroxylation or dehydration of the major mineralogical phases generated during Portland cement hydration or the pozzolanic reaction (C-S-H gel, ettringite, C_4_AH_13_, and C_4_AcH_11_). The quantification of mass loss related to hydrated phases (HP) of the pastes at 28 days are presented in Table 5.

Through Table 5, it is possible to observe that the P-NS paste presented higher HP compared to the P-REF paste. However, in some studies, it is possible to observe that due to the high reactivity of NS, a layer of C-S-H is formed on the surface of the clinker, impairing its complete hydration after 28 days and, consequently, the mechanical performance of the cementitious composites [9,10]. The P-NSF4 and P-NSF8 pastes resulted in HP similar to that of the P-NS paste and higher than that of the P-REF paste.

#### 3.2.3. Compressive Strength

Figure 5 presents the compressive strength results at ages 2, 7, and 28 days for Portland cement pastes.

It is observed that at 2 days, the P-NS presented 10% higher compressive strength in relation to the P-REF. This behavior was expected due to the effect of NS in the initial ages. As the APTES content increased in the NSFs, there was a decrease in the mechanical strength of the pastes after 2 days. In the case of the NSF8, the strength was zero, indicating that the paste did not even set. This behavior has already been observed in cement pastes with NSFs with a high APTES content [32]. Both the effect of APTES and the superplasticizer can influence this behavior. Attention is drawn to the behavior of P-NSF4, which already at 2 days showed 5% higher compressive strength compared to the reference and only 5% less compressive strength than NS pastes. After 7 days, a recovery of the compressive strength of pastes with NSF is observed. Even P-NSF8, which showed zero strength after 2 days, presented compressive strength very close to the reference mortar after 7 days. It is observed that P-NSF4 presented compressive strength about 10% higher than P-REF and higher than P-NS. At 28 days, all pastes with NSF showed higher compressive strength than P-REF and also than P-NS. Both P-NSF4 and P-NSF8 showed about a 20% increase in compressive strength compared to P-REF and a 10% increase in compressive strength compared to P-NS. This improvement can be understood with the improvement in the hydration of the cement grains caused by the delay of the reactions. According to Rupasinghe et al. [39], this behavior of extending the period of hydration reactions promotes an improvement in the hydration of the cement grain. Liu et al. [24] also adds that this reduction in the speed of hydration reactions is linked to the formation of high-density C-S-H on the surface of the cement grain at early ages.

To verify whether the strength differences were significant at each age, a statistical analysis was performed using the one-way ANOVA, considering a significance level of 0.05. At each assessed age, a *p*-value lower than the significance level was observed, that is, there was a significant difference between the means. Thus, the Duncan test was performed to classify the samples into heterogeneous groups of compressive strength, as shown in Table 6.

At 2 days of hydration, four strength groups were formed, with group 1 having the lowest strength and group 4 having the highest strength. At this age, the P-NSF4 showed a lower strength than the P-NS, but higher than the P-REF. On the other hand, the highest APTES content did not make it possible to obtain results from the P-NSF8 paste, which was classified in group 1 (without strength).

At 7 days of hydration, three groups were formed, with group 1 having the lowest strength and group 3 having the highest strength. The P-NSF4 sample showed statistically equal strength to the P-NS paste, both being higher than the P-REF paste. At 2 days, the P-NSF8 paste had the lowest strength.

At 28 days of hydration, only two strength groups were formed, with group 1 having the lowest strength and group 2 having the highest strength. The P-NSF4 and P-NSF8 samples were classified in group 2, with no significant difference between them. As the P-NS paste was classified in groups 1 and 2, the NSF provided a trend of increased strength in relation to the P-NS. The P-REF paste was classified in group 1, being the one with the least strength. In this case, the P-NS paste showed a tendency to increase the mechanical performance at 28 days of hydration compared to the P-REF. This demonstrates that NSF is more efficient to promote an increase in mechanical performance than NS in cementitious composites between 7 and 28 days of hydration.

#### 3.2.4. Mercury Intrusion Porosimetry

The mercury intrusion porosimetry test aimed to evaluate the porous structure of the P-REF, P-NS, P-NSF4, and P-NSF8 pastes and to evaluate their pore distribution. Figure 6 shows the volume of mercury intruded in Portland cement pastes after 28 days.

In general, it was observed that there was a refinement of the porous structure of the P-NS, P-NSF4, and P-NSF8 pastes in relation to the P-REF paste due to the displacement of the peak to the left in the graph. The P-NS results are in line with the literature [40]. According to the literature, this refinement of the porous structure is attributed to the pozzolanic activity of NS which, even in small amounts, exerts significant influences on the microstructure of cement-based materials. Puentes et al. [41] noted, through mercury intrusion porosimetry, that the addition of 1% nanosilica reduced the total porosity of cement paste samples by about 40% compared to the reference. It is also observed that even with the delay in hydration reactions, P-NSF4 and P-NSF8 showed intense pozzolanic activity after the induction period, which contributed to the refinement of the pores.

Observing Table 7, it is noteworthy that the P-NS paste had a smaller average pore diameter than the P-REF paste due to the pozzolanic activity of the NS which consumes the CH from the Portland cement hydration reactions to form added C-S-H, corroborating the literature [40].

The P-NSF 4 sample showed an average pore diameter even smaller than that of the P-NS paste, indicating that the functionalization of NS with a low APTES content improves the hydration of Portland cement in relation to the P-NS paste. The P-NSF8 sample, despite the delay in cement hydration, showed an average pore size similar to the sample with NS. Other information that can be extracted from Table 5 is related to the sizes of the capillary pores, which are responsible for reducing the mechanical performance and durability of the pastes [42]. The P-NS, P-NSF4, and P-NSF8 pastes had fewer large capillary pores than the REF sample. The P-NS and P-NSF4 pastes were the ones that resulted in the smallest total volume of mercury intruded in the range of pores with sizes related to the large capillaries. The P-NSF 4 paste was the sample that showed the highest pore refinement, indicating that at this age, in terms of durability, these pastes may have been the ones with the best performance. The increase in compressive strength at 28 days of P-NSF4 and P-NSF8 may also be related to the decrease in large and medium capillaries in these samples. The NSF samples caused alterations in the microstructure of the Portland cement pastes, which show a probable efficiency in the cement hydration process, despite the delay presented. This efficiency was more pronounced in the NSF 4 sample that showed compressive strength at 2 days close to P-REF and P-NS (showing that the effect of increasing the induction time did not affect the strength gain at these ages) and development of higher strength to P-NS samples at advanced age (28 days).

#### 3.2.5. Computed Tomography

The macroporosity of cement pastes was studied using computed axial tomography (CAT) on 2.5 cm high samples, and the results are shown in Figure 7 and Table 8. Using the CAT technique, it was possible to analyze the pore volume of each paste within the following classification: 1. pores with volume > 0.5 mm^3^; 2. pores with a volume between 0.5 mm^3^ and 0.1 mm^3^; 3. pores with a volume between 0.1 mm^3^ and 0.05 mm^3^; and 4. pores with a volume between 0.05 mm^3^ and 0.01 mm^3^. It was also possible to determine the average pore volume of each paste within this pore volume range.

It is observed that the P-REF paste had a higher mean pore volume (0.031 mm^3^) than the NSF4 (0.022 mm^3^) and NS (0.026 mm^3^) pastes. The pozzolanic effect of NS was probably effective, in these two cases, to reduce the mean pore volume within the pore volume range between those higher than 0.5 mm^3^ and 0.01 mm^3^. The NSF8 paste had the highest average pore volume (0.047 mm^3^). It is also observed that the NSF8 paste had the highest percentage of pores with a volume higher than 0.5 mm^3^ and the lowest percentage of pores with a more refined volume (between 0.05 mm^3^ and 0.01 mm^3^).

A possible justification for this behavior is related to the amount of superplasticizer additive required in the P-NS, P-NSF4, and P-NSF8 pastes to reach standardized consistency. It can be observed in Table 6 that as the superplasticizer additive content increased, the average pore volume for the analyzed paste and the percentage of pores with a volume higher than 0.5 mm^3^ increased. As the superplasticizer additive content increases, there may be an increase in the air content incorporated in the paste, which justifies this behavior.

These results indicate an increase in the complexity of the porous structure of pastes with NSFs. By the mercury intrusion porosimetry test, a refinement of the pores in the micrometric scale is observed, probably due to the pozzolanic reaction and improvement in the cement hydration process in these pastes, while the increase in the superplasticizer additive content in these pastes in relation to the reference paste can increase the pore volume on the macrometric scale.

## 4. Conclusions

This research evaluated the microstructure of Portland cement pastes with nanosilica functionalized with different proportions of aminosilane for use in high-performance concretes. The APTES/NS volume ratios used for NS functionalization were 4:60 (NSF 4) and 8:60 (NSF 8). The research conclusions are presented below:The method used for functionalization was effective in grafting aminosilane functional groups at different levels on the surface of the studied nanosilicas.The microstructural analysis techniques proved that the aminosilane groups were grafted and that there was a change in the NSF characterization as the functionalization content changed. This result was pointed out by DLS (with an increase in hydrodynamic radius), TG (increase in mass loss), and FTIR (accentuated bands and changes in OH bonds).Sample P-NSF 4 had a smaller amount of additive to reach the established consistency index in relation to P-NS. The P-NSF 8 sample required a larger amount of additive. For this sample, the increase in the functionalization ratio led to the adsorption of more water within the NSF 8 which led to the need for a higher SP content than the P-NS.The NSF’s initially acted as setting retardants within the cementitious composites, with the induction period being longer in samples with NSF with a higher proportion of aminosilane (NSF 8). NSF 4 had the shortest induction period compared to NSF8. After the induction period, the cement hydration reactions were similar to the NS sample in terms of heat development.The extension of the induction period provoked in the P-NSF4 and P-NSF8 pastes a delay in the gain of strength in initial ages, but with recovery of strength after 7 days. The P-NSF 4 sample showed compressive strength at 2 days superior to the P-REF sample and close to P-NS. Samples P-NSF4 and P-NSF8 showed compressive strength higher than sample P-REF and higher than P-NS at 28 days.Through the TGA test, we observed that the samples with NSF with different levels of amine group altered the hydration kinetics, as evidenced by the CH content related to the delay of the hydration and pozzolanic reactions, as well as the hydrated phases of the pastes at 28 days of hydration.Porosimetry by mercury intrusion showed that the use of NSF4 and NSF8 caused pore refinement similar or even superior to the P-NS sample. The P-NSF4 paste had the lowest accumulated intruded volume, indicating a smaller volume of voids compared to the other pastes. On the other hand, the computed tomography test observed an increase in the macropositivity of the pastes with NSF8, probably due to the higher incorporation of air due to the larger amount of superplasticizer additive incorporated in these pastes.

Based on these conclusions, we have that the variation in the proportion of functionalizations of NS with aminosilane caused changes mainly in terms of hydration kinetics, pore size distribution, and the compressive strength of Portland cement pastes. In this sense, it is concluded that the functionalization of NS with different levels of aminosilane can produce cementitious materials with specific properties for the production of high-performance concrete.

## Figures and Tables

**Figure 1 materials-16-05675-f001:**
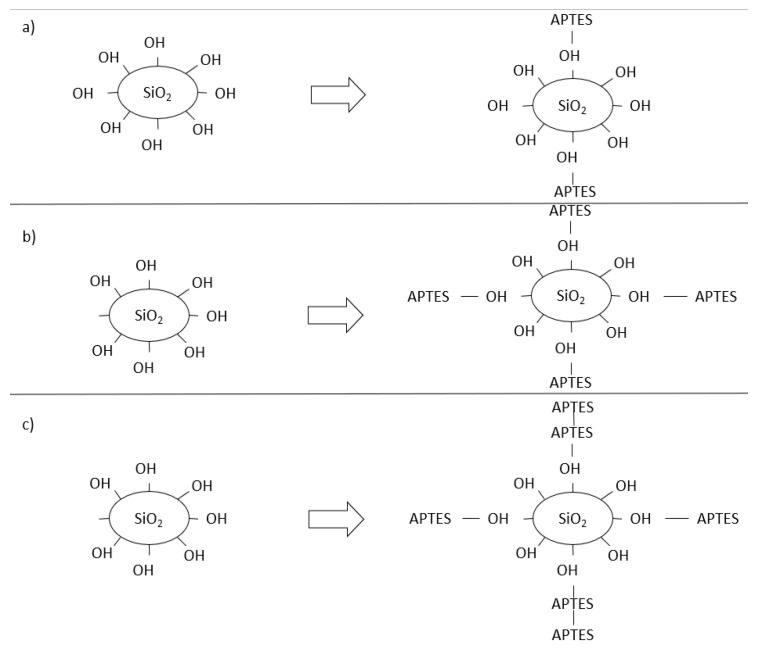
(**a**) NS and (**b**) NSF partially saturated surface; (**c**) NSF surface saturated with excess of APTES.

**Figure 2 materials-16-05675-f002:**
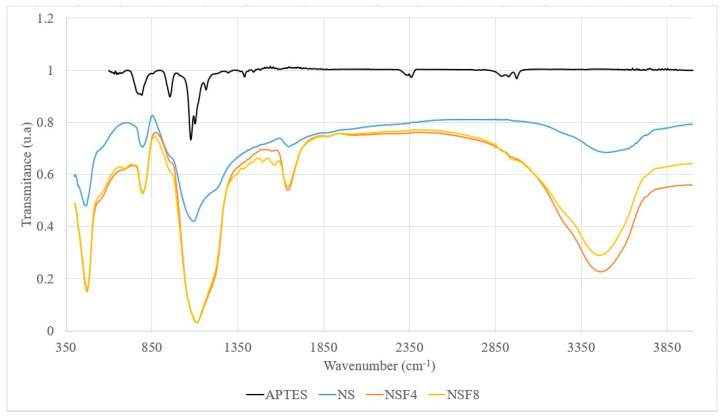
FTIR spectra of NS, NSF4, NSF8, and APTES samples.

**Figure 3 materials-16-05675-f003:**
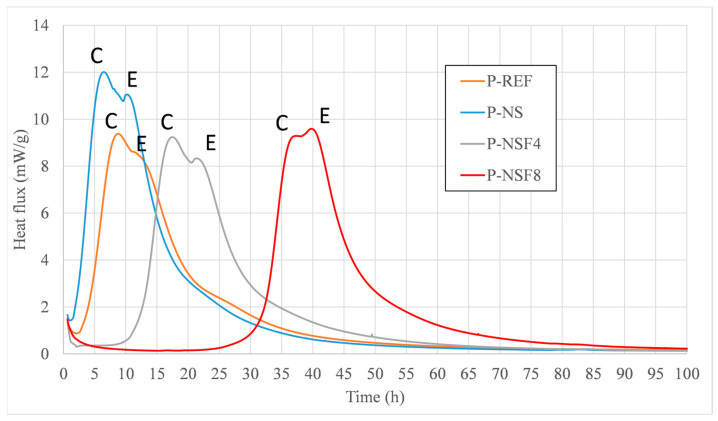
Heat flux released from the P-REF, P-NS, P-NSF4, and P-NSF8 pastes.

**Figure 4 materials-16-05675-f004:**
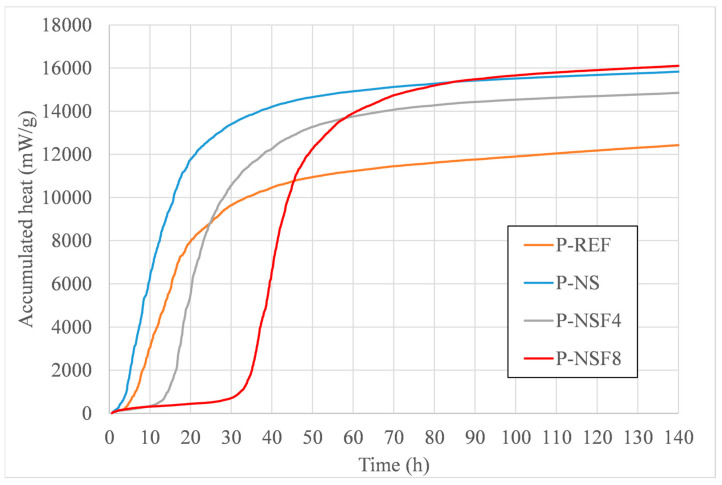
Accumulated released heat from the P-REF, P-NS, P-NSF4, and P-NSF8 pastes.

**Figure 5 materials-16-05675-f005:**
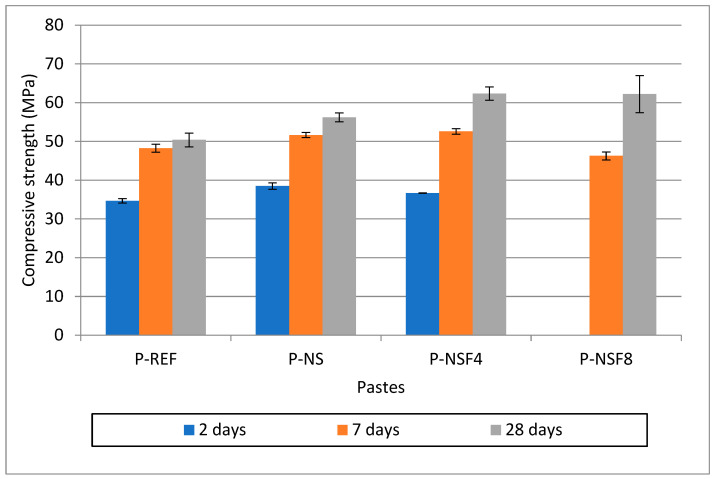
Compressive strength of the pastes at 2, 7, and 28 days.

**Figure 6 materials-16-05675-f006:**
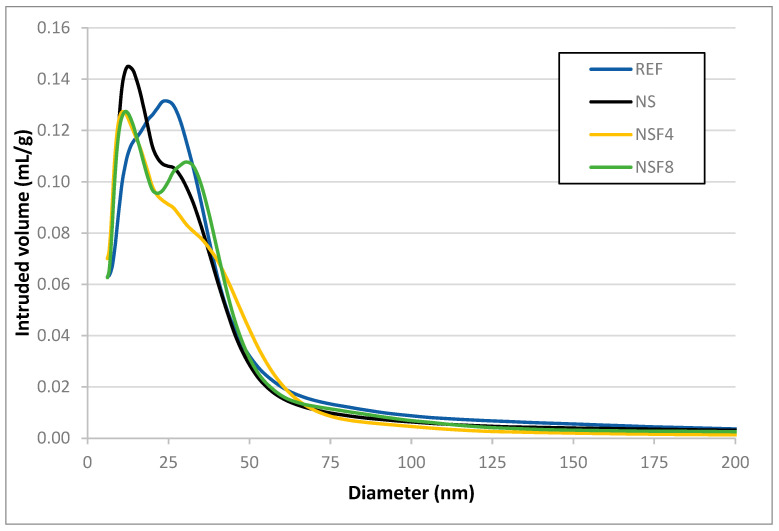
Volume of mercury intruded in the pastes at 28 days.

**Figure 7 materials-16-05675-f007:**
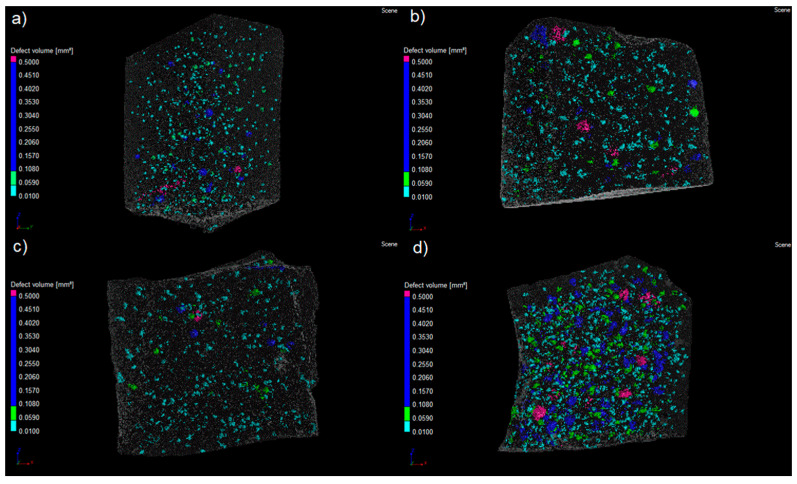
Porosity distribution of the samples by computed tomography analysis of the pastes after 28 days of hydration: (**a**) Reference, (**b**) NS, (**c**) NSF4, (**d**) NSF8.

**Table 1 materials-16-05675-t001:** Composition of cement pastes.

Paste	Material						Mini Slump (mm)
	Cement (g)	NS or NSF		Superplasticising Additive		Water (g)	
		Solids (g)	Aqueous Suspension (g)	(g)	(%) by Mass of Cement		
P-REF	2000.00	0.00	0.00	5.60	0.28	695.94	119.72
P-NS	1980.00	20.00	61.48	16.00	0.80	646.91	114.28
P-NSF4	1980.00	20.00	128.87	14.00	0.70	580.97	116.76
P-NSF8	1980.00	20.00	137.93	25.00	1.25	563.93	114.23

Note 1: The amount of water in the colloidal nanosilica (functionalized or not) and in the superplasticizer additive was subtracted from the amount of mixing water for all pastes, resulting in the water/solids ratio being = 0.35. Note 2: Material (solids content)—NS (32.53%), NSF4 (15.52%), NSF8 (14.50%), superplasticizer additive (27.44%).

**Table 2 materials-16-05675-t002:** Physical and chemical characterizations of NS and NSF samples.

Sample	Color	Specific Mass (g/cm^3^)	pH	Loss of Ignition (%)	Solid Content (%)
NS	Transparent	1.23	10.44	5.40	32.53
NSF4	White with transparency	1.19	10.05	8.56	15.52
NSF8	Opaque white	1.17	9.52	16.61	14.01

**Table 3 materials-16-05675-t003:** Hydrodynamic diameter and content of functionalized amine groups in nanosilica.

Sample		TGA	DLS	
	Total Mass Loss (%)	Functionalization Content (%)	Hydrodynamic Diameter (nm)	Size Distribution Area (%)
NS	5.19	0.00	21.74	100.00
NSF4	8.41	3.22	26.04 (A1)	95.10
			3516.00 (A2)	4.90
NSF8	14.76	9.57	203.50 (A1)	39.30
			858.80 (A2)	56.70

**Table 4 materials-16-05675-t004:** CH content and CH index of the pastes at 2, 7, and 28 days of hydration.

Age	Content		Paste		
		P-REF	P-NS	P-NSF4	P-NSF8
2 days	CH content in relation to the total mass of the sample	10.43	9.63	11.11	6.84
	CH index in relation to REF	100.0%	92.3%	106.5%	65.6%
7 days	CH content in relation to the total mass of the sample	11.48	9.42	11.19	8.94
	CH index in relation to REF	100.0%	82.1%	97.5%	77.9%
28 days	CH content in relation to the total mass of the sample	12.16	9.33	11.13	9.88
	CH index in relation to REF	100.0%	76.7%	91.5%	81.3%

**Table 5 materials-16-05675-t005:** Mass loss related to HP of the pastes at 28 days of hydration.

Paste	Mass Loss in the 50 °C to 400 °C Range (%)
P-REF	6.66
P-NS	7.93
P-NSF4	7.80
P-NSF8	7.97

**Table 6 materials-16-05675-t006:** Statistical analysis of the compressive strength of the pastes at 2, 7, and 28 days of hydration.

Paste	Compressive Strength (MPa)	Standard Deviation (MPa)	*p*-Value		Group		
2 days				Group 1	Group 2	Group 3	Group 4
P-REF	34.7	0.57	0.0000		X		
P-NS	38.5	0.85					X
P-NSF4	36.7	0.06				X	
P-NSF8	0.0	0.00		X			
7 days				Group 1	Group 2		Group 3
P-REF	48.2	1.03	0.0001		X		
P-NS	51.6	0.66					X
P-NSF4	52.6	0.70					X
P-NSF8	46.3	1.03		X			
28 days				Group 1		Group 2	
P-REF	50.4	1.77	0.0055	X			
P-NS	56.2	1.13		X		X	
P-NSF4	62.4	1.72				X	
P-NSF8	62.2	4.80				X	

Note: Groups are age-independent. In this way, the groups of one age are not related to the groups of other ages.

**Table 7 materials-16-05675-t007:** Pore characteristics of pastes at 28 days.

Paste	Total Porosity (%)	Average Pore Diameter (nm)	Intruded Mercury Volume (mL/g)	
			Large Capillary (10.000–50 nm)	Medium Capillary (50–10 nm)
P-REF	18.72	16.7	0.00910	0.07590
P-NS	18.20	15.4	0.00670	0.07840
P-NSF4	17.61	14.9	0.00650	0.07070
P-NSF8	18.07	15.7	0.00800	0.07390

**Table 8 materials-16-05675-t008:** Porosity of pastes after 28 days of hydration by computed tomography.

Pastes		P-REF	P-NS	P-NSF4	P-NSF8
Average pore volume (mm^3^)		0.031	0.026	0.022	0.047
Pore volume distribution (%)	>0.5 mm^3^	0.17	0.42	0.22	0.68
	0.5–0.1 mm^3^	4.36	2.06	1.97	8.22
	0.1–0.05 mm^3^	6.18	4.64	3.94	11.75
	0.05–0.01 mm^3^	89.34	92.86	93.85	79.34
Superplasticizer additive content (% cement mass)		0.28%	0.80%	0.70%	1.25%

## Data Availability

Not applicable.

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
