# Peer review of "Analysis of the Microstructure and Porosity of Cement Pastes with Functionalized Nanosilica with Different Contents of Aminosilane"

_materials, 2023, doi:10.3390/ma16165675_

Round 1
Reviewer 1 Report
The present study deals with the effect of aminosilane content in functionalized nanosilica (NSF) on hydration and physical-mechanical properties of Portland cement pastes. The results are clearly presented and they are scientifically commented by authors. Only a few comments could be taken into account by authors:
- The composition of pastes is better to be expressed with a way so they would be reproducible by others. Additionally, change “(%) relative to cement mass” to maybe “% by mass of cement” or any appropriate phrase authors want.
- line 185: what are the conditions of the chamber (temperature, humidity).
Author Response
Attached letter-1

Reviewer 2 Report
Congratulations on an original work.
1- Researchers have one word different studies with this title “ Effect of amine functionalized nanosilica on the cement, hydration and on the physical-mechanical properties, J Nanopart Res (2020) 22: 234 of Portland cement pastes”. The title of this article should be changed.
2- After writing the APTES, NS, CH, TGA expansion in the Abstract part, its abbreviation should be made.
3- The abbreviation in not correct “ nano cementitious material (NMCS)”
4- The material and method part of the article should be rewritten. Experimental results and discussions should be given in the result and discussion section of the article.
5- Result part should be named Result and Discussion.
6- The articles i have mentioned below can be used about citation. “Ilker Ustabas , Sakir Erdogdu, Ihsan Omur,and Erol Yilmaz, Pozzolanic Effect on the Hydration Heat of Cements Incorporating Fly Ash, Obsidian, and Slag Additives, Hindawi Advances in Civil Engineering Volume 8 OCT 2021, Article ID 2342896, 12 pages https://doi.org/10.1155/2021/2342896,”
Ilker Ustabas, Ayberk Kaya, Comparing the pozzolanic activity properties of obsidian to those of fly ash and blast furnace slag, Construction and Building Materials 164 (2018) 297–307.
7- The conclusion part of the article should be rewritten. The expressions especially the numerical interpretation in the result and discussion section should not be used again in the conclusion part of the article.
The English of the entire article should be checked.
Author Response
Attached letter-2

Reviewer 3 Report
Can you complete the paper with SEM images of the particle size?
I recommend that you write table 6 according to the writing instructions of the journal.
I recommend you to write lines 343-345, without italics.
Author Response
Attached letter-3

Round 2
Reviewer 1 Report
Authors made changes and I thus recommend the paper to be accepted for publication.